# Increased Fruit and Vegetable Consumption Mitigates Oxidative Damage and Associated Inflammatory Response in Obese Subjects Independent of Body Weight Change

**DOI:** 10.3390/nu15071638

**Published:** 2023-03-28

**Authors:** Salah Gariballa, Ghada S. M. Al-Bluwi, Javed Yasin

**Affiliations:** Internal Medicine, College of Medicine & Health Sciences, United Arab Emirates University, Al-Ain P.O. Box 15551, United Arab Emirates

**Keywords:** fruits and vegetables, obesity, oxidative damage, inflammation

## Abstract

Introduction: The growing prevalence of obesity and related type 2 diabetes is reaching epidemic proportions in the Gulf countries. Oxidative damage and inflammation are possible mechanisms linking obesity to diabetes and other related complications, including cardiovascular disease (CVD). Aims: To measure the effects of increased fruit and vegetable consumption on body weight, waist circumference, oxidative damage, and inflammatory markers. Materials and Methods: We recruited and followed up with 965 community free-living subjects. All recruited subjects had fruit and vegetable intakes, physical activity, antioxidants, and markers of oxidative damage and inflammation measured at baseline and follow up. A validated, semi-quantitative food-frequency questionnaire was used to assess subjects’ fruit and vegetable consumption. We stratified subjects based on their daily fruit and vegetable consumption and compared metabolic risk factors between those with high fruit and vegetable consumption and those with low consumption. A multiple logistic regression analysis was performed to determine the independent effects of fruit and vegetable intake on changes in body weight and waist circumference (WC). Results: A total of 965 community free-living subjects (801 (83%) females, mean (SD) age 39 ± 12 years) were recruited and followed up with for a mean (SD) period of 427 ± 223 days. Using WHO cut-off points for body mass index (BMI), 284 (30%) subjects were overweight and 584 (62%) obese, compared to 69 (8%) at normal body weight. An increased fruit and vegetable consumption was associated with a significant decrease in inflammatory markers (hs CRP, TNF-α) and oxidative damage markers (TBARs) and with increased antioxidant enzymes (catalase, glutathione peroxidase) compared to a low consumption (*p* < 0.05). The benefits of an increased fruit and vegetable consumption in obese subjects was independent of changes in body weight and WC and was maintained at follow up. Conclusion: Our results support the beneficial role of a higher fruit and vegetable intake in obese subjects independent of changes in body weight and WC.

## 1. Introduction

More than 2.1 billion people in the world, around 30% of the global population, are overweight or obese, with this figure set to rise to almost half of the world’s adult population by 2030 [1]. Most of the obesity epidemic is concentrated in the developing world. For example, in the Gulf region, the prevalence of obesity and related pathologies, including diabetes and other cardiovascular disease (CVD) risk factors, is increasing rapidly and reaching an epidemic proportion [1,2,3]. The United Arab Emirates (UAE), in particular, has one of the highest prevalences of obesity-related diabetes mellitus in the world [3]. Visceral (abdominal) obesity is common and more closely related to morbidity, especially in the Middle East, which includes the UAE. Recent studies have reported that the waist-to-hip ratio showed the strongest relationship with CVD worldwide [4,5]. If a raised waist–hip ratio were to be used to assess the risk of cardiovascular disease, the proportion of people classified as obese worldwide would increase substantially, especially in the Middle East and South East Asia [4]. Possible mechanisms that relate visceral obesity to an increased risk of diabetes and other related complications include inflammation and oxidative damage [6,7]. The antioxidants found in fruits and vegetables might promote health by combating oxidative damage resulting from increased free radicals, which are linked to the pathogenesis of many chronic diseases, including obesity, diabetes, and CVD [8]. Furthermore, fruits and vegetables may prevent weight gain and help weight loss through their high dietary fiber and low energy content. We have recently reported that an increased adiposity is associated with an increased inflammation and a decrease in antioxidants (found in fruits and vegetables) in obese subjects in the UAE [9].

Previous narrative and systematic reviews on the relationship between fruit and vegetable consumption and obesity outcomes produced conflicting results [10,11]. Another systematic review on the relationship between fruit and vegetable intake and adiposity reported that the inverse relationship between the two among overweight adults appears weak [12]. The authors concluded that additional research is needed to clarify the nature of and mechanisms for the effects of fruit and vegetable consumption on adiposity [12]. Because little is known about strategies that specifically reduce visceral obesity, we investigated the effects of an increased fruit and vegetable consumption on metabolic risk factors, including oxidative damage, inflammatory markers, body weight, and waist circumference.

## 2. Material and Methods

A convenient sample of community free-living subjects from hospitals and primary health care clinics in the city of Al Ain in the UAE, serving a total population of 600,000, were approached through the local press and invited to take part in this study. Following the receipt of informed written consent and their recruitment to the study, eligible subjects had clinical, dietary, physical activity anthropometric measurements, and a fasting 10 mL of blood taken for the measurement of antioxidants, markers of oxidative damage, inflammation, and other related clinical, nutritional, and biochemical variables at baseline. Individuals who had severe chronic clinical or psychiatric diseases or were participating in other intervention trials, on dietary supplements, taking anti-obesity medications, or unable to give informed written consent were excluded. The local research ethics committee of Al Ain Medical District has approved the study.

### 2.1. Measurements

All participants received a baseline clinical assessment to collect their demographic and medical data, history of chronic illness, and smoke, alcohol, and drug intake. Anthropometric data, including body weight, height, and body mass index (BMI), were measured using the Tanita body composition analyzer, with which the results are shown on an easy-to-read display screen and printed on a sheet. Waist circumference was measured to the nearest 0.1 cm using a flexible plastic tape at the mid-point between lower ribs and iliac crest.

Fruit and vegetable intake: A validated, short semi-quantitative food-frequency questionnaire designed for self-administration following a brief verbal discussion was used to assess the subjects’ fruit and vegetable intake. Participants reported the typical frequency of their consumption of particular food items during the previous 12 months, and responses were used to assess the average weekly nutrient consumption of each individual. The full version of this questionnaire was developed and validated against a 7-day weighted dietary intake. It has also been compared with numerous other diets and used in many other studies [13]. Calorie intake: Calorie intake was measured for a subgroup of subjects using locally validated 24-h recalls, once at the baseline assessment and once at the follow-up visit. Physical activity: A validated questionnaire was used to assess participants’ occupational and leisure-related physical activity. Data were obtained on the frequency and duration of daily or weekly physical activity sessions lasting 20 min or more, in which subjects became breathless or sweaty.

Blood samples: Details on the measurement of metabolic risk factors have been previously published [14]. Briefly, fasting blood samples were drawn and stored in 2 vacutainer tubes containing potassium EDTA as an anticoagulant. The samples were thoroughly mixed at room temperature and immediately transferred to the laboratory. Both tubes were centrifuged immediately for 10 min at 4000 rotations/min. Plasma and serum were collected and stored at −80 °C for the future determination of biochemical outcome measurements. Antioxidants: Commercially available Cayman’s colorimetric assay kits from the USA (kit numbers 706002, 707002, and 703102) were used to measure antioxidant enzymes, including glutathione (GSH), superoxide dismutase (SOD), glutathione peroxidase (GPx), and catalase. We used commercially available enzyme-linked immunosorbent assay (ELISA) methods to measure plasma TNF. Lipid Peroxidation: Concentrations of the lipid-peroxidation product Thiobarbituric Acid Reactive Substances (TBARS) was measured using an assay kit (no. 10009055) from the Cayman Chemical Company, 1180 E. Ellsworth Rd., Ann Arbor, MI 48108, USA. Protein Oxidation: The content of protein-bound carbonyls, used to assess the extent of protein oxidation, was determined calorimetrically using a reagent kit (10005020) from the Cayman Chemical Company. Circulating levels of renal and liver functions, lipids, and high-sensitivity C-reactive protein (hsCRP) were measured using an automated analyzer Integra 400 Plus (Roche Diagnostics, Mannheim, Germany).

### 2.2. Statistics and Analysis

Fruit and vegetable intake was divided into low consumption (≤median serving/day) and high consumption (>median serving/day). In addition, fruit and vegetable intake was further divided into 4 equal quartiles. Both the one-way and two-way ANOVA tests and the nonparametric Kruskal–Wallis H-test were used to identify within- and between-group differences, and a *p* value of <0.05 was considered significant. The effects of changes in diet and lifestyle and other prognostic indices on changes in body weight and waist circumference were identified at follow up. A multiple logistic regression analysis was used to examine the influence of fruit and vegetable consumption and other prognostic indicators on changes in body weight and waist circumference at follow up. The outcome variable (weight or WC) was dichotomized (i.e., loss vs. no change or gain) and then analyzed as a dependent variable against a number of independent variables including age, occupation, level of education, baseline status, and changes in fruit and vegetable consumption and physical activity. The relationship between fruit and vegetable consumption, the presence or absence of obesity, and a decrease in body weight and WC are presented graphically using the Kaplan–Meier survival curve with the probability of body weight or WC loss at the time of follow up taken as the point of interest.

## 3. Results

A total of 965 subjects (801 (83%) females, mean (SD) age 39 ± 12 years) are included in the final analysis. Using WHO cut-off points for BMI, 284 (30%) subjects had a high health risk (overweight) (BMI ≥ 25–29.9), and 584 (62%) subjects had an increased health risk (obese) (BMI ≥ 30) compared to 69 (8%) at a normal health risk (BMI ≤ 25) (Table 1). Table 1 shows the baseline characteristics, demographics, physical activity, anthropometrics, blood pressure, and biochemical makeup of subjects, stratified by the BMI cut-off points for classifying someone as overweight or obese. Subjects with a high health risk (BMI ≥ 30) were significantly older and had an increased prevalence of hypertension compared to those with a satisfactory health risk (Table 1). Waist circumferences, inflammatory markers and glycemic control markers were significantly higher and HDL lower in people with a high health risk or an increased health risk compared to those with a satisfactory health risk (*p* < 0.05).

Table 2 and Table 3 show the baseline and follow-up anthropometrics, BP, lipid profile, and inflammatory and oxidative damage markers according to fruit and vegetable intake, divided into four quartiles. The levels of oxidative damage and inflammatory markers were significantly lower and antioxidant enzymes significantly higher in subjects at the fourth quartile of fruit and vegetable consumption, relative to those at the first quartile of the distribution (*p* < 0.05). Similar results of decreased oxidative and inflammatory markers and increased antioxidant enzymes are seen in subjects with a high fruit and vegetable consumption (above median serving/day) compared to those with a low consumption both at baseline and follow up (Table 4 and Table 5). Glycemic control markers, both glucose and HbA1c, were lower in those with a high fruit and vegetable consumption compared to those with a low consumption; however, these results did not reach statistical significance. There were no statistically significant differences in daily calorie intake between those with a high fruit and vegetable consumption compared to those with a low consumption either at baseline or follow up (*p* > 0.05).

Results of the multiple logistic regression analyses of the influence of some clinical prognostic variables on changes in the body weight and WC (loss vs. gain) of the study population at follow-up are shown in Table 6 and Table 7. Only baseline physical activity, but not increased fruit and vegetable consumption, showed an independent significant association with weight loss. Figure 1 and Figure 2 show positive associations between a high fruit and vegetable consumption and a decrease in both body weight and WC; however, this did not reach statistical significance. Similarly, Figure 3 and Figure 4 show an association between obesity diagnoses (BMI ≥ 30) and levels of fruit and vegetable consumption, which also did not reach statistical significance.

As part of the physical activity assessment, subjects were categorized based on their activity during leisure time into very, moderately, or not physically active. Only 164 (16%) of the study participants with complete data reported being very active during leisure time at baseline compared to 75 (7%) at follow up.

## 4. Discussion

In this study, we found that an increased fruit and vegetable consumption with or without an increase in physical activity mitigates inflammatory responses and oxidative damage independent of changes in body weight and/or WC circumference in obese Emirati subjects. A higher fruit and vegetable consumption may also have a weak but positive effect on glycemic control.

The benefits of an increased fruit and vegetable consumption in obese subjects were independent of changes in body weight and WC and were maintained at follow up.

These findings may have important public health implications in light of the fact that approximately around 35% of lost weight in obese subjects is regained one year following treatment, and 50% of patients will return to their baseline weight by the fifth year after weight loss [15]. Reflecting the worst-case scenario, roughly 90 percent of people who lose a lot of weight eventually regain just about all of it. In a meta-analysis of 29 long-term weight loss studies, more than half of the lost weight was regained within two years, and more than 80% within five years [16].

With the growing epidemic of general and visceral obesity and its related complications, including diabetes and CVS disease, in our society and other similar nations, there is an urgent need for a simple and practical intervention to help reduce this burden. Diet, for example, has been reported by the Global Burden of Disease study to contribute more to diseases such as obesity, diabetes, and CVD than physical inactivity, smoking, and alcohol combined [17]. Furthermore, adherence to a diet with a relatively large number of fruits and vegetables has been shown to reduce the chance of chronic disease and significantly decrease chances of CVS and total mortality [18]. It has been estimated that increasing the population’s consumption of fruits and vegetables by one portion per day and net consumption by two servings a week would prevent 5.2 million deaths from CVS disease globally within just one year [19]. To combat the obesity epidemic, focusing on increasing fruit and vegetable intake rather than counting calories has also been recommended based on good evidence [20].

### 4.1. Fruit and Vegetable-Mechanisms of Action

These results give more credence to the recently launched program embraced by cities such as New York, dubbed the “Fruit and Vegetables Prescription Program”, which allows doctors to “prescribe” fresh fruit and vegetables to overweight or obese patients by giving them “health bucks” that are redeemable at local farmers’ markets [21].

Fruits and vegetables are rich in water, fiber, vitamins (including antioxidants), and minerals. They help weight loss through their low energy content and high dietary fiber content [22]. Fiber-rich foods tend to be more satiating as a result of their relatively low energy density compared to low-fiber foods. Additionally, dietary fiber, especially soluble fiber, could increase the viscosity of a diet and slow down digestion, which stimulates the release of gut hormones such as cholecystokinin and glucagon-like peptide 1 and promotes satiety [23]. Moreover, the slower digestion and absorption rate of carbohydrate would lead to a reduced postprandial blood glucose response, which over time could improve insulin sensitivity and influence fuel partitioning to favor fat oxidation [22].

### 4.2. Antioxidants, Oxidative Damage, and Inflammation

Possible mechanisms that relate obesity to an increased risk of diabetes include inflammation and oxidative damage. In obese patients, subclinical inflammation has been found to correlate with markers of oxidative stress in their adipose tissue, and this may be the mechanism behind obesity-related metabolic syndrome, insulin resistance, and diabetes mellitus. Furthermore, both oxidative stress and low-grade inflammation may be causatively linked to the development, progression, and complication of diabetes in obese patients [24,25]. Oxidative stress, defined as an imbalance between the generation of free oxygen radicals and the antioxidant defense system, results from an increased production of reactive oxygen species known to trigger cytotoxic reactions that are damaging to membrane lipids, proteins, nucleic acids, and carbohydrates. A number of studies have revealed a link between oxidative stress, obesity, diabetes, and other related complications. Recent research does indeed support a close link between oxidative stress and the evolution of diabetes, revealing that oxidative stress occurs before the clinical manifestation of late diabetic complications, which suggests that it plays a key role in the pathogenesis of the disease [24]. In addition, a number of studies have reported an association between oxidative stress and insulin resistance and that some antioxidants may improve insulin resistance [25]. A recent systematic review and meta-analysis has reported that increasing the daily intake of green, leafy vegetables could significantly reduce the risk of type 2 diabetes [26]. Other relatively recent data also suggest that dietary antioxidant intake may be a predictor of risk level for the development of features of metabolic syndrome, such as adiposity, or impairments in systolic blood pressure, serum glucose, and free fatty acids, as well as some inflammatory biomarkers in healthy subjects [27]. Although the role of both oxidative stress and low-grade inflammation in the development, progression, and complication of diabetes in obese patients is well accepted, the benefits of an increased fruit and vegetable consumption in the prevention and treatment of visceral obesity-related complications, including type 2 diabetes mellitus, has not been sufficiently studied [20,22]. Our study results suggest that a higher fruit and vegetable intake plays a beneficial role in subjects with visceral obesity, mediated by a decreased inflammatory response and the mitigation of oxidative damage.

### 4.3. Effects of Fruit and Vegetable Consumption on Weight and WC Loss

Our study findings suggest no statistically significant association between fruit and vegetable intake and changes in body weight or WC. In contrast, other studies have reported that a higher consumption of red/purple fruits and vegetables, such as tomatoes, red onions, dates, red cabbage, watermelons, cherries, red grapes, berries, strawberries, and red plums, has been associated with lower abdominal fat gain [28]. Another study reported the beneficial effects of a higher intake of fruit, vegetables, and cereal fiber on abdominal obesity prevention [23]. The effects of fruits and vegetables on visceral obesity could be mediated by a decreased inflammatory response, the mitigation of oxidative damage associated with inflammatory cytokines that favor lipolysis, and lipid oxidation instead of fat storage [7,29,30]. This is clearly an area for further research.

### 4.4. Limitations and Strength of the Study

Our study has some limitations, including the use of a self-administered semi-quantitative food-frequency questionnaire and 24-h recalls to assess a subject’s fruit, vegetable, and calorie intakes, respectively. The bias associated with dietary intake is a well-known limitation of food-frequency questionnaires and 24-h recalls compared to observed intake in adults [31]. Another limitation is the non-random selection of the study population. Nevertheless, we have adjusted for important prognostic differences, such as age, gender, education, marital status, and physical activity in the analysis. Body composition measurements, for example, were performed digitally and printed on a sheet with little room for observer error. Biochemical analyses were also carried out by a laboratory technician not involved in the recruitment or data collection.

## 5. Conclusions

We have demonstrated in this study that a higher fruit and vegetable consumption mitigates inflammatory responses and oxidative damage independent of changes in body weight in a population with the highest rates of obesity and related diabetes in the world. A higher fruit and vegetable consumption may also have a weak but positive effect on glycemic control. In light of obese people having difficulties losing weight and easily regaining lost weight, coupled with the availability of healthy food choices, these findings could have enormous public health implications by mitigating obesity-related adverse health effects in our community and worldwide. Urgent actions are needed to counteract obesity in the Gulf countries, including the UAE, and worldwide, such as reducing fat, sugar, and salt in manufactured products and enabling easier and cheaper access to healthy food, namely fruits and vegetables, coupled with reducing commercial pressure on people (particularly children) to consume high-energy products.

## Figures and Tables

**Figure 1 nutrients-15-01638-f001:**
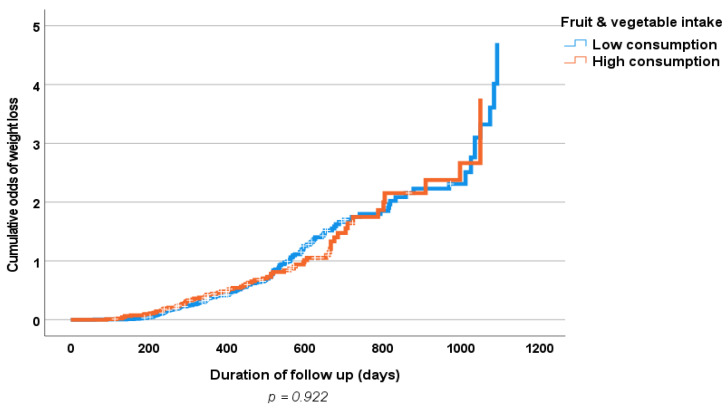
Kaplan-Meier curve of effects of follow-up fruit and vegetable consumption (high consumption > 3.7 servings/day vs. low consumption ≤ 3.7 servings/day) on body weight (loss vs. gain).

**Figure 2 nutrients-15-01638-f002:**
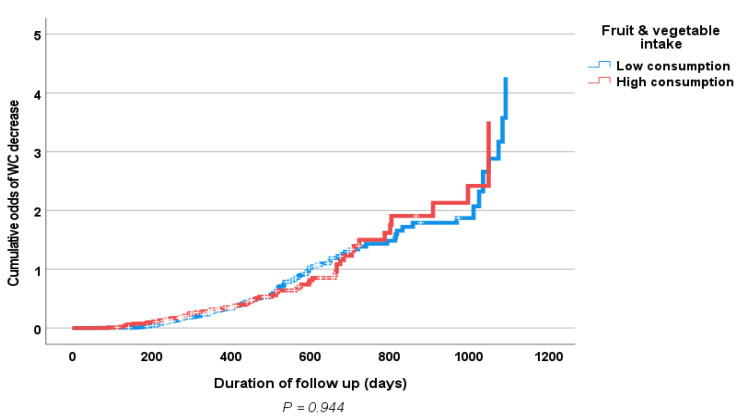
Kaplan–Meier curve of effects of follow-up fruit and vegetable consumption (high consumption > 3.7 servings/day vs. low consumption ≤ 3.7 servings/day) on waist circumference (loss vs. gain).

**Figure 3 nutrients-15-01638-f003:**
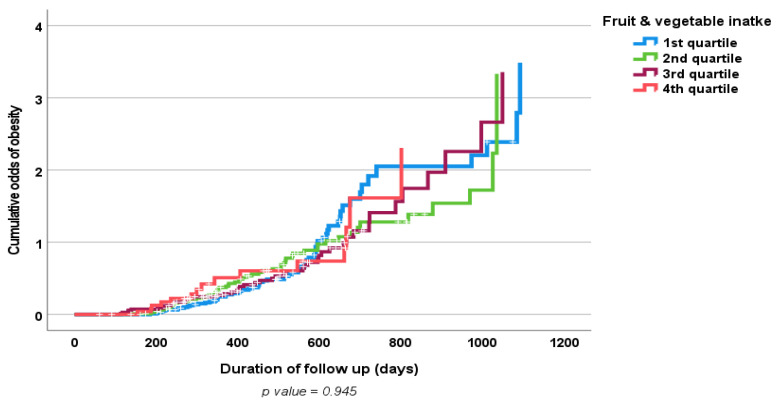
Kaplan–Meier curve of odds of obesity diagnosis according to the fruit and vegetable consumption quartiles at follow up.

**Figure 4 nutrients-15-01638-f004:**
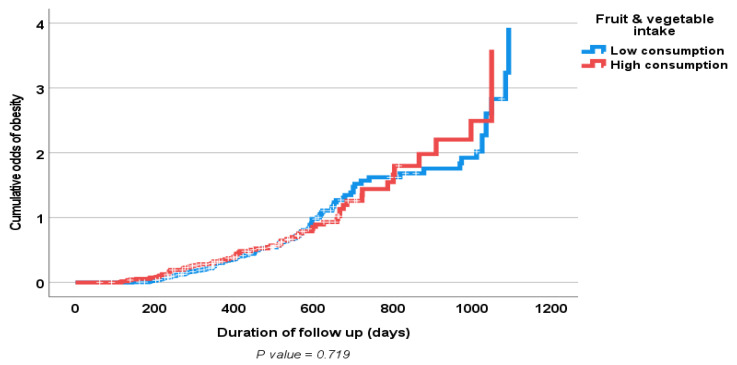
Kaplan–Meier curve of odds of obesity diagnosis according to fruit and vegetable consumption (high consumption > 3.7 servings/day vs. low consumption ≤ 3.6 servings/day) at follow up.

**Table 1 nutrients-15-01638-t001:** Baseline characteristics and metabolic risk factors according to body mass index (BMI) health-risk categories, using the WHO BMI cut-off points, mean (SD).

	BMI			
	Normal Risk (BMI ≤ 25) (*n* = 69)	High Health Risk (BMI 25.1–29.9) (*n* = 284)	Increased Health Risk (BMI ≥ 30) (*n* = 584)	*p* Value
Age (years)	34 (12)	39 (12)	40 (12)	0.001
Females, *n* (%)	53 (77)	167 (59)	491 (84)	0.001
Previous diabetes, *n*(%)	6 (9)	39 (14)	86 (15)	0.390
Previous hypertension, *n*(%)	4 (6)	31 (11)	104 (18)	0.001
Waist circumference (cm)	81 (11)	90 (8)	103 (13)	0.001
Systolic BP (mmHg)	117 (9)	121 (15)	123 (14)	0.001
Diastolic BP (mmHg)	74 (6)	74 (8)	76 (10)	0.051
Total cholesterol (mmol/L)	4.7 (1)	5.0 (0.9)	4.9 (0.9)	0.073
Triglycerides (mmol/L)	1.1 (0.6)	1.38 (1)	1.42 (1)	0.482
HDL (mmol/L)	1.39 (0.4)	1.21 (0.4)	1.10 (0.3)	0.048
HbA1c, = (%)	5.4 (0.6)	5.9 (1)	6.0 (1)	0.001
Glucose (mmol/L)	5.8 (2)	8.8 (10)	8.9 (11)	0.060

Numbers sometimes do not add up because of missing values.

**Table 2 nutrients-15-01638-t002:** Baseline anthropometrics, BP, lipid contents, and inflammatory and oxidative damage markers according to fruit and vegetable consumption quartiles, mean (SD).

	1st Quartile (0.5 to 3.0 Servings/Day) (*n* = 159)	2nd Quartile (3.5 to 4.5 Servings/Day) (*n* = 177)	3rd Quartile (4.6–5.6 Servings/Day) (*n* = 189)	4th Quartile (5.7 to 9.8 Servings/Day) (*n* = 166)	*p* Value *
Body weight (kg)	85.6 (17)	86.5 (16)	86.5 (16)	86.2 (15)	0.947
Body mass index	34.3 (5)	33.3 (6)	33.0 (6)	33.3 (6)	0.165
Waist circumference (cm)	101 (14)	99.5 (13)	99.9 (14)	98.2 (12)	0.359
Systolic BP (mmHg)	121 (14)	119 (14)	123 (14)	122 (13)	0.080
Diastolic BP (mmHg)	74 (9)	72 (9)	76 (11)	77 (10)	0.001
HbA1c (%)	6.0 (1)	5.8 (0.7)	6.1 (1.4)	6.0 (1.5)	0.715
Glucose (mmol/L)	11.6 (16)	9.3 (9)	11.2 (14)	6.8 (4)	0.059
hs-CRP (mg/L)	8.1 (8)	5.7 (7)	4.2 (5)	4.1 (4)	0.001
TNFα (pg/mL)	13.7 (4.2)	11.9 (5.7)	7.7 (5)	6.4 (4.7)	0.001
Glutathione (GSH) (nM/mL)	6.3 (4.1)	6.0 (3.5)	7.3 (5.6)	7.5 (3)	0.418
Superoxide dismutase (U/mL)	4.4 (3)	3.5 (1.5)	2.9 (1.3)	2.5 (0.9)	0.001
Catalase (nmol/min/mL)	33 (20)	46 (32)	62 (33)	71 (33)	0.001
Glutathione peroxidase (ng/mL)	64 (102)	100 (89)	158 (81)	206 (71)	0.001
TBARS (nmol/mL)	30 (13)	27 (15)	18 (12)	18 (13)	0.001
Protein carbonyl (nmol/mL)	95 (51)	93 (64)	121 (63)	149 (73)	0.001

* *p* value for difference between fruit and vegetable consumption quartiles using one-way ANOVA.

**Table 3 nutrients-15-01638-t003:** Follow-up anthropometrics, BP, lipid contents, and inflammatory and oxidative damage markers according to fruit and vegetable consumption quartiles, mean (SD).

	1st Quartile (0.5 to 2.8 Servings/Day) (*n* = 116)	2nd Quartile (2.9 to 3.8 Servings/Day) (*n* = 99)	3rd Quartile (3.9–4.9 Servings/Day) (*n* = 113)	4th Quartile (5.0 to 9.8 Servings/Day) (*n* = 101)	*p* Value *
Body weight (kg)	82 (15)	83.5 (17)	77.5 (12)	82 (19)	0.108
Body mass index	32.4 (5)	33.3 (7)	31.5 (4)	32.3 (5)	0.250
Waist circumference (cm)	97 (13)	97 (16)	94 (13)	94 (13)	0.521
Systolic BP (mmHg)	118 (11)	119 (10)	119 (11)	118 (9)	0.819
Diastolic BP (mmHg)	73 (9)	72 (7)	72 (8)	71 (8)	0.578
HbA1c (%)	6.1 (1.4)	5.9 (1.1)	6.0 (1.4)	5.7 (0.9)	0.318
Glucose (mmol/L)	6.8 (6)	6.2 (5)	5.8 (4)	6.5 (4)	0.488
hs-CRP (mg/L)	6.0 (6)	6.5 (8)	4.1 (5.5)	3.5 (3.4)	0.011
TNFα (pg/mL)	7.8 (3.7)	7.7 (4.1)	6.4 (4.4)	5.1 (2.9)	0.001
Glutathione (GSH) (nM/mL)	5.6 (3.4)	6.7 (4)	6.8 (3.6)	5.8 (3.3)	0.512
Superoxide dismutase (U/mL)	4.8 (3.5)	4.8 (3.4)	3.7 (3.3)	3.1 (1.4)	0.003
Catalase (nmol/min/mL)	49 (19)	51 (25)	68 (32)	72 (29)	0.001
Glutathione peroxidase (ng/mL)	52 (21)	76 (43)	116 (100)	126 (49)	0.001
TBARS (nmol/mL)	31 (11)	27 (11)	21 (10)	20 (12)	0.001
Protein carbonyl (nmol/mL)	73 (55)	88 (68)	111 (86)	137 (72)	0.001

* *p* value for difference between fruit and vegetable consumption quartiles using one-way ANOVA.

**Table 4 nutrients-15-01638-t004:** Baseline anthropometrics, BP, lipid and antioxidant contents, and markers of oxidative damage and inflammation in those with a high fruit and vegetable consumption vs. a low consumption.

Mean (SE)	Low Consumption (≤4.4 Servings/Day)	High Consumption (>4.4 Servings/Day)	Two-Sided *p* Values *
	(*n* = 374)	(*n* = 295)	
Calorie intake (Kal/day)	1085 (631)	1274 (716)	0.191
Body weight (kg)	86 (17)	88.6 (17)	0.670
Body mass index	33.8 (6)	33.0 (5)	0.111
Waist circumference (cm)	100 (13)	99 (13)	0.179
Systolic BP (mmHg)	120 (13)	122 (14)	0.064
Diastolic BP (mmHg)	73 (14)	77 (11)	0.001
HbA1c (%)	5.9 (0.8)	6.0 (1.5)	0.346
Glucose (mmol/L)	10.6 (14)	9.0 (9)	0.220
hs-CRP (mg/L)	6.7 (7)	4.1 (4)	0.001
TNFα (pg/mL)	12.5 (5)	7.0 (5)	0.001
Glutathione (GSH) (nM/mL)	6.2 (4)	7.8 (5)	0.036
Superoxide dismutase (U/mL)	3.8 (2.3)	2.7 (1.2)	0.001
Catalase (nmol/min/mL)	41 (28)	67 (33)	0.001
Glutathione peroxidase (ng/mL)	87 (95)	181 (80)	0.001
TBARS (nmol/mL)	28 (14)	18 (12)	0.001
Protein carbonyl (nmol/mL)	98 (61)	133 (70)	0.001

* *p* value for difference in anthropometrics, BP, lipid and antioxidant contents, and markers of oxidative damage and inflammation between those with a high fruit and vegetable consumption and those with a low consumption.

**Table 5 nutrients-15-01638-t005:** Follow-up anthropometrics, BP, lipid and antioxidant contents, and markers of oxidative damage and inflammation in those with a high fruit and vegetable consumption vs. a low consumption.

Mean (SE)	Low Consumption (≤3.7 Servings/Day)	High Consumption (>3.7 Servings/Day)	Two-Sided *p* Values *
	(*n* = 200)	(*n* = 229)	
Calorie intake (Kal/day)	1155 (735)	1149 (634)	0.966
Body weight (kg)	82.9 (16)	79 (15)	0.047
Body mass index	32.9 (6)	31.7 (4)	0.055
Waist circumference (cm)	96.9 (14)	93.6 (13)	0.054
Systolic BP (mmHg)	118 (10)	119 (10)	0.196
Diastolic BP (mmHg)	73 (8)	72 (8)	0.693
HbA1c (%)	6.1 (1.3)	5.8 (1.1)	0.093
Glucose (mmol/L)	6.8 (5)	6.0 (4)	0.165
hs-CRP (mg/L)	6.7 (7)	3.8 (4)	0.001
TNFα (pg/mL)	8.0 (4)	5.7 (3)	0.001
Glutathione (GSH) (nM/mL)	5.9 (4)	6.6 (3.6)	0.350
Superoxide dismutase (U/mL)	4.66 (3.4)	3.56 (2.4)	0.014
Catalase (nmol/min/mL)	49 (19)	68 (31)	0.001
Glutathione peroxidase (ng/mL)	60 (33)	119 (75)	0.001
TBARS (nmol/mL)	30 (11)	21 (11)	0.001
Protein carbonyl (nmol/mL)	78 (59)	124 (80)	0.001

* *p* value for difference in anthropometrics, BP, lipid and antioxidant contents, and markers of oxidative damage and inflammation between those with a high fruit and vegetable consumption and those with a low consumption.

**Table 6 nutrients-15-01638-t006:** Multiple logistic regression analysis of the influence of some clinical prognostic variables on change in body weight (loss vs. gain) of study population at follow-up.

Variable	Regression Coefficient	Standard Error	Odd Ratio for Unit Change (95% CI)	*p* Value
Age (years)	−0.020	0.018	0.980 (0.947–1.014)	0.249
Sex (male/female)	−0.248	0.574	0.780 (0.253–2.405)	0.666
Marital status (married, unmarried, divorced)	0.311	0.207	1.364 (0.909–2.047)	0.133
Level of education (Primary, secondary, graduate)	−0.226	0.136	0.797 (611–1.040)	0.095
Baseline physical activity (Not active, moderately active, very active)	0.867	0.403	2.380 (1.08–5.245)	0.032
Difference between baseline and follow up physical activity	−0.564	0.376	0.569 (0.272–1.189)	0.134
Baseline fruit and vegetable consumption (servings/day)	−0.011	0.023	0.989 (0.946–1.034)	0.638
Difference between baseline and follow-up fruit and vegetable consumption	0.007	0.020	1.007 (0.969–1.047)	0.727

**Table 7 nutrients-15-01638-t007:** Multiple logistic regression analysis of the influence of some clinical prognostic variables on changes in the waist circumference (loss vs. gain) of the study population at follow up.

Variable	Regression Coefficient	Standard Error	Odd Ratio for Unit Change (95% CI)	*p* Value
Age (years)	0.007	0.015	1.007 (0.978–1.036)	0.636
Sex (male/female)	0.282	0.531	1.326 (0.468–3.757)	0.596
Marital status (married, unmarried, divorced)	0.017	0.165	1.017 (0.736–1.407)	0.917
Level of education (primary, secondary, graduate)	−0.056	0.112	0.946 (0.760–1.177)	0.616
Baseline physical activity (not active, moderately active, very active)	0.365	0.304	1.440 (0.793–2.616)	0.231
Difference between baseline and follow-up physical activity	−0.372	0.286	0.689 (0.393–1.207)	0.193
Baseline fruit and vegetable consumption (servings/day)	−0.013	0.019	0.987 (0.952–1.024)	0.495
Difference between baseline and follow-up fruit and vegetable consumption	0.010	0.016	1.010 (0.979–1.042)	0.540

## Data Availability

Data is available upon request to the corresponding author.

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
