# Peer review of "Increased Fruit and Vegetable Consumption Mitigates Oxidative Damage and Associated Inflammatory Response in Obese Subjects Independent of Body Weight Change"

_nutrients, 2023, doi:10.3390/nu15071638_

Round 1

Reviewer 1 Report

The authors demonstrated in this study that higher fruits and vegetables consumption mitigates the inflammatory response and oxidative damage independent of body weight change in a population with the highest rates of obesity and related diabetes in the world. I have some questions about this manuscript.

1.       Lines 15 and 23: 963 or 965?

2.       Women accounted for 83% of the survey population, and overweight and obese individuals accounted for 92%. How can the sample's representativeness be ensured?

3.       A total of 965 subjects [801 (83%) females, mean (SD) age 39±12 years] are included in the final analysis. ” However, Table 1 only shows data for 53+161+261 females. Could you please provide an explanation for this discrepancy?

4.       Table 1: The data formatting and presentation lack organization, and it is advisable to use n(%) and mean ± SD.

5.       Tables 4 and 5: Calorie intake (Kal/day)? hs-CRP (mg/l) α; TNFα (pg/ml) α? TBARS (nmol/ml) + ; Protein carobonyl (nmol/ml) +? Could you kindly provide an explanatory note for the symbols α and +? Additionally, please indicate the method of data presentation.

6.       Tables 6 and 7 depict the outcomes of multiple logistic regression analysis; however, displaying the results of multi-classified variables is not appropriate. Modifications are recommended.

7.       Line 210: “only 64 (16%) of study population reported being very active at leisure times at baseline compared to 75 (7%) at follow up.” How are these two percentages calculated?

8.       The author's follow-up seems to have no effect. It appears that only two cross-sectional studies were conducted.

9.       The author solely ascertained that the consumption of fruits and vegetables was correlated with certain markers of inflammation and oxidative damage in the single factor analysis, the conclusion “increased fruits and vegetables consumption mitigates inflammatory response and oxidative damage” does not seem to be adequately supported.

10.   The limitations of the study were not stated completely.

WRITING

11.   There are minor issues throughout that shouldn’t occur. Some illustrative examples:

The font size in the chart title is inconsistent, and the format of P value is inconsistent.

Table 4: “Protein  carobonyl”

Line 303: “marital status sand physical activity”

Author Response

Reviewer 1 

1.Lines 15 and 23: 963 or 965?

Correct number of subjects recruited provided-965

2.Women accounted for 83% of the survey population, and overweight and obese individuals accounted for 92%. How can the sample's representativeness be ensured?

       Subjects included in this research are a convenient sample of community free living people approached and agreed to take part in the survey.  Looking at the age strata and the rate of overweight and obesity compared with previous published data, the recruited sample however they do seems to represent the local population. This point is acknowledged under the Methods (line 68) and also part of study Limitations (line 311). 

3.A total of 965 subjects [801 (83%) females, mean (SD) age 39±12 years] are included in the final analysis. ” However, Table 1 only shows data for 53+161+261 females. Could you please provide an explanation for this discrepancy?

      Numbers (%) of females in table corrected (line 147)

4.Table 1: The data formatting and presentation lack organization, and it is advisable to use n(%) and mean ± SD.

Mean (SD) or n(%)accurately stated opposite relevant variables in all Tables

5.Tables 4 and 5: Calorie intake (Kal/day)? hs-CRP (mg/l) α; TNFα (pg/ml) α? TBARS (nmol/ml) + ; Protein carobonyl (nmol/ml) +? Could you kindly provide an explanatory note for the symbols α and +? Additionally, please indicate the method of data presentation.

      Symbols removed and methods of data analysis provided as footnotes (Table 4, line 178; Table 5, line 185)

6.Tables 6 and 7 depict the outcomes of multiple logistic regression analysis; however, displaying the results of multi-classified variables is not appropriate. Modifications are recommended.

      As we have clearly explained under ‘statistical analysis’ lines 123-128, we used multiple logistic regression analysis to see which of the following factors age, sex, marital status, level of education, physical activity and fruits and vegetables consumption are predictive of weight or waist circumference loss.  The explanatory variables and the dependent variable were coded and aanalysed using the method described by Douglas Altman in his text book ‘Practical Statistics for Medical Research, Chapman & Hall, P 351-358”.

7.Line 210: “only 64 (16%) of study population reported being very active at leisure times at baseline compared to 75 (7%) at follow up.” How are these two percentages calculated?

      New text provided to clarify the above point (lines 217 & 2018).

8.The author's follow-up seems to have no effect. It appears that only two cross-sectional studies were conducted.

     Our study is a longitudinal prospective cohort study.  The study design and the follow up assessment gave us the opportunity to improve the precision of measurements, confirm the associations observed at baseline but more importantly investigate factors predictive and associated with weight or waist circumference loss including fruits and vegetables consumption.

9.The author solely ascertained that the consumption of fruits and vegetables was correlated with certain markers of inflammation and oxidative damage in the single factor analysis, the conclusion “increased fruits and vegetables consumption mitigates inflammatory response and oxidative damage” does not seem to be adequately supported.

      Although we agree that we cannot infer causality between increased fruits and vegetables consumption and obesity related pathology we can however from the data and analysis presented conclude that that increased fruits and vegetables consumption mitigates inflammatory response and oxidative damage.  More importantly Reviewer 2 in complete agreement with our conclusion, quote “The data are adequate to support the conclusions”

10.The limitations of the study were not stated completely.11.   There are minor issues throughout that shouldn’t occur. Some illustrative examples:

New text provided on another limitation of the study (line 312)

The font size in the chart title is inconsistent, and the format of P value is inconsistent.

Table 4: “Protein  carobonyl”

Line 303: “marital status sand physical activity”

All minor editorial issues including the ones above addressed

Reviewer 2 Report

Review Manuscript ID: nutrients-2287989 titled “Increased fruits and vegetables consumption mitigate oxidative damage and associated inflammatory response in obese subjects independent of body weight change” by Gariballa et al.

The manuscript reports data on the effects of fruits and vegetables consumption on body weight, waist circumference, oxidative damage and inflammatory markers. A total of 963 community free-living subjects were recruited and followed on an average for 427 days. Main findings were that increased fruits and vegetables consumptions were associated with significant decrease in inflammatory markers (hs CRP, TNF-α), oxidative damage markers (TBARs) and increased antioxidant enzymes (catalase, glutathione peroxidase) compared to those with low consumption.

The manuscript is well-written, well-structured, and easy to read. The data are adequate to support the conclusions. The findings are rather novel and the manuscript represents a contribution to the literature. I have no major criticism, only some minor comments;

Please check table 2 and 3, there seems to be typos in the headings.

Please check table 6 and 7, the text style in not consistent.

The figures did not add much relevant information according in my opinion.

Page 13, line 316 “Guld” should probably be “Gulf”. There are also some minor grammatical errors – please review the manuscript from a language point of view.

Author Response

Reviewer 2

  1. Please check table 2 and 3, there seems to be typos in the headings.

Typos addressed

  1. Please check table 6 and 7, the text style in not consistent.

Text style made consistent in all tables.

  1. The figures did not add much relevant information according in my opinion.

The figures more powerfully visually illustrate the relationship albeit non-significant between high and low fruits and vegetables consumptions and the odds of general and abdominal obesity diagnosis at follow up.

  1. Page 13, line 316 “Guld” should probably be “Gulf”. There are also some minor grammatical errors – please review the manuscript from a language point of view.

 All typos corrected
